# The Fermentation Law of Biogenic Amines in the Pre-Fermentation Process Is Revealed by Correlation Analysis

**DOI:** 10.3390/foods14040583

**Published:** 2025-02-10

**Authors:** Lijing Liu, Jinyu Zhao, Dapeng Lu, Jiao Zhao, Guqing Duan, Tong Zhu, Yongjin Hu

**Affiliations:** 1College of Food Science and Technology, Yunnan Agricultural University, Fengyuan Road 452, Kunming 650201, China; lilyliutt@163.com (L.L.); 15825027015@163.com (D.L.); qingguduan159@sina.com (G.D.); tongzhu@ynau.edu.cn (T.Z.); 2Yunnan Coffee Modern Industry College, Yunnan Agricultural University, Fengyun Road 452, Kunming 650201, China

**Keywords:** Mouding sufu, pre-fermentation, biogenic amines, physicochemical components, texture

## Abstract

Sufu is a traditional Chinese condiment with a distinctive flavor. The umami characteristics are primarily attributed to the hydrolysis of proteins, which produces amino acids and biogenic amines (BAs). Excessive levels of BAs can pose health risks, leading to adverse effects, such as headaches, digestive disorders, and abnormal blood pressure. However, the mechanisms leading to the formation of BAs in fermented bean curd remain insufficiently explored. To explore this phenomenon, an analysis was conducted on the texture, physicochemical properties, and BA content dynamic changes of sufu, fermented with *Mucor racemosus*, at different intervals, by high-performance liquid chromatography-mass spectrometry (HPLC-MS). During the fermentation process, the total biogenic amines exhibited a downward trend, with levels decreasing from 60.66 to 38.19 mg/kg. Spermidine, spermine, and cadaverine were identified as the main biogenic amines produced during the fermentation process. At 0 h and 24 h, spermidine levels significantly decreased (*p* < 0.05), but no significant differences were observed at 48 h and 72 h. At 96 h, spermidine levels significantly decreased again (*p* < 0.05). There was a positive relationship between the bioamines and water content and a negative correlation with soluble protein levels. Additionally, low pH inhibits the formation of BAs, while a soft texture was found to increase their production. The results of this study also confirmed the mechanism of BAs formation. These findings offer valuable insights into the safety and quality control of Mouding sufu by clarifying the BA dynamics during pre-fermentation.

## 1. Introduction

Sufu, often referred to as “Chinese cheese”, is a traditional Chinese fermented soybean product with distinct cultural characteristics [1]. Famous fungal-fermented bean curds in China include Guangxi Guilin sufu, Beijing Red Square sufu, Zhejiang Shaoxing sufu, and Guangdong Yangjiang sufu. Due to its subtropical production environment and unique pre-fermentation process, Mouding sufu is one of the most representative products, holding the honorary title of the National Geographic Indication product in Yunnan. Mouding sufu has a high moisture content and a soft, delicate texture. It is renowned for its rich nutritional value, unique flavor, and affordability, and it is widely loved by the public as an appetizer [2,3], especially in the Yunnan province. In addition, fermented soy products are known to enhance nutrients such as soy isoflavones, antioxidants, and active substances [4,5].

Mouding sufu has a history of over 1000 years, and is produced in Mouding County, Yunnan Province, at an altitude of approximately 1490 m, with an average temperature ranging from 14 °C to 25 °C. The mild and humid climate of Mouding County provides an ideal environment for the natural fermentation of Mouding sufu, which is praised by food experts in China as “the sole softness of bean curd”, due to its high total water content and its distinctive flavor. The production of the fermented bean curd follows the traditional Chinese technology; it is made from soybeans using a process that includes grinding, making sufu [6], pre-fermentation, seasoning, marinating, bottling, and post-fermentation [1,7]. *Mucor racemosus* appears to be the most widely employed strain for sufu production by the manufacturers in Mouding. According to traditional sufu-fermentation techniques, the production process comprises pre-fermentation and post-fermentation stages. During the pre-fermentation process, tofu is inoculated with *Mucor racemosus* on the surface and cultivated for 96 h, until the formation of furry tofu blocks occurs. Subsequently, the prepared furry tofu block is dried for approximately 12 h before being cured for 2–3 days. In the end, the salt–furry tofu block is carefully placed in a jar with a dressing mixture for about 3–6 months, which is called the post-fermentation process.

The content of biogenic amines (BAs), hardness, and smoothness are important indicators for evaluating the quality and acceptability of fermented bean curd, as well as key factors in determining the maturity stage during the production process. Recent research on Mouding sufu has primarily focused on process improvement, strain optimization, functional nutrition, and safety evaluation. Process improvement efforts aim to enhance the quality of fermented bean curd by adjusting fermentation conditions, such as temperature and salinity, or by incorporating functional ingredients like sour whey and starter cultures [8]. Strain optimization emphasizes the screening and application of specific strains, such as Bacillus licheniformis and Aspergillus flavus, to improve the flavor, nutritional value, and safety of fermented bean curd [9]. Additionally, research on functional nutrition has concentrated on the extraction and analysis of metabolites in sufu, including amino acids, peptides, and antioxidants [6]. However, studies on the formation mechanisms of BAs and their relationship with the physicochemical structure of sufu remain relatively limited, highlighting an important direction for current research [10]. For instance, Cheng et al. [11] investigated the biochemical changes related to protein degradation in low-temperature fermented bean curd, suggesting that Bacillus flavus could serve as a substitute strain for production. Furthermore, Xie et al. [12] demonstrated that the use of mixed starters and α-ketoglutaric acid could enhance the taste quality and color of low-salt fermented bean curd. Overall, while significant advancements have been made in understanding the production and enhancement of Mouding sufu, further exploration into the safety aspects, particularly regarding BAs, is essential for ensuring product quality and consumer safety.

This study systematically elucidates the formation mechanism of BAs and their correlation with physicochemical indicators during pre-fermentation stage of Mouding sufu. It investigates key physicochemical properties such as moisture content, water-soluble proteins, total acidity and amino acid nitrogen, while tracking and detecting the types and levels of BAs throughout pre-fermentation. By analyzing the correlation between these physicochemical indicators and the accumulation of BAs, the research identifies the primary factors influencing their production and explores the biosynthesis mechanisms involved. Furthermore, an in-depth analysis of the relationship between BAs and quality attributes, including texture and nutritional value, provides preliminary theoretical insights into regulating BAs. These findings aim to enhance product quality and ensure safety in the subsequent production stages, offering valuable guidance for the industrial production of fermented bean curd.

## 2. Materials and Methods

### 2.1. Sufu Preparation Process

Select high-quality soybeans that are plump and free of insect damage and mold. After cleaning, soak the soybeans for 8 h to 18 h until they are completely softened. Use a grinder to process the soybeans into a slurry, ensuring that the okara is fine and free of whole soybeans. Rapidly heat the slurry, maintaining its temperature between 92 °C and 98 °C. At 85 °C, add Mouding sour slurry water to coagulate the mixture, and maintain the temperature in a water bath at 85 °C for 20 min. Then, press the mixture into molds for 1 h to 2 h. Arrange the tofu blocks evenly in a clean container, ensuring they do not stick together. Keep the fermentation room temperature between 20 °C and 25 °C until the surface of the tofu blocks is covered with upright mycelium [7]; this stage is known as the pre-fermentation stage and typically takes four days. Subsequently, sun-dry the sufu blocks until their moisture content decreases from 40% to 30%. Wash the surface of the tofu blocks with a saline solution to remove spores, mycelium, and some enzymes. Mix the cleaned tofu blocks with salt, chili powder, and spices. Place the mixed tofu blocks into fermentation containers, stored for around 180 days, this stage is referred to as the post-fermentation stage. Figure 1 illustrates the critical control points of the sufu pre-fermentation process.

### 2.2. Preparation of Mold Spore Suspension

Measure an appropriate amount of mold liquid medium into an Erlenmeyer flask, according to the required volume of spore suspension. Add a proportional amount of deionized water, seal the flask, and sterilize it in an autoclave at 120 °C for 20 min. Using a sterilized and cooled inoculating loop, transfer *Mucor racemosus* stolonifer from a PDA (Potato Dextrose Agar) plate into a sterilized mold medium, performing this step in a laminar flow hood. Place the inoculated Erlenmeyer flask in an incubator shaker at 25 °C for 48 h to activate the culture. The cultured mycelium should be white, robust, and free of contaminants. Transfer the cultured medium into centrifuge tubes and centrifuge for 10 min. Decant the supernatant, add sterilized saline, and shake thoroughly to form a spore suspension. Adjust the concentration of the spore suspension to 10^7^ CFU/mL using the turbidimetric method. Prepare the spore suspension fresh, for immediate use, and do not store it.

### 2.3. Sample Preparation and Collection

Select 200 g of plump, mold-free soybeans per batch (one batch equals one tofu block). Soak the soybeans in an appropriate amount of deionized water at room temperature, for 12 h to 18 h. After soaking, grind the soybeans with water, at a ratio of 1:7, for about 10 min, until smooth soy milk is obtained, free of whole soybeans. Filter out the okara and boil the soy milk. Cool it to 85 °C and add Mouding sour slurry water at a 15% volume ratio. Mix well and place in a water bath at 85 °C for 20 min. Press the mixture into tofu molds and cut into blocks to obtain the tofu blocks.

Inoculate the tofu blocks by immersing all six sides evenly in a mold spore suspension. Arrange the tofu blocks neatly in fermentation boxes, leaving some space around each block to allow for *Mucor racemosus* growth, which promotes better fermentation [7]. Place the fermentation boxes in a constant temperature and humidity incubator, maintaining a relative humidity of 75% to 85% and a temperature of 20 °C to 25 °C for 4 d. At each fermentation time point (0, 24, 48, 72, 96 h), collect 7 samples, tracking three batches in parallel. One sample from each time point is used for texture analysis and one is used for moisture content measurement. These two indices are measured immediately after sampling. The remaining five samples are stored in an ultra-low temperature freezer (−80 °C) for further analysis of other physicochemical indices and BAs.

### 2.4. Determination of Moisture Content and pH

The moisture content of Mouding sufu was determined using the “direct drying method”, in accordance with the Chinese National Standard [13]. The pH value was measured using a pH meter (Jingke Leici Instruments, Shanghai, China), following the pH measurement method specified in the Chinese National Standard [14]. A cooled 5.0 g of the sample was tested in a 50 mL centrifuge tube, after adding 20 mL of deionized water to fully grind the sample, or after using a homogenizer to disperse and homogenize the sample, centrifuging the sample at 5000 r/min for 20 min, and then transferring the supernatant to a 50 mL volumetric flask, determining the volume with deionized water, pouring out 30 mL, and then measuring the sample directly with a calibrated pH meter. Each sample was analyzed 3 times, following the same process.

### 2.5. Determination of Water-Soluble Protein, Amino Acid Nitrogen and Total Acid

The water-soluble protein content was determined by following the recommended standard [15] issued by the Ministry of Agriculture of China (2016). Amino acid nitrogen content was quantified using the acid-based titration method. The total acid content in sufu was measured in accordance with the “acidity meter method”, as specified in the Chinese National Standard [16]. Each of the above measurements was performed in triplicate to ensure accuracy and reproducibility.

### 2.6. Texture Measurement

The texture was determined by the TPA method [17], in which the curd samples were cut into small squares, with flat surfaces and uniform sizes, and the texture profile of the curd was measured by using the texture analyzer probe with the second downward pressure. The specific parameters of the Texture Profile Analysis (TPA) were as follows: probe model: P/36R, pre-test speed 2.0 mm/s, mid-test speed 1.0 mm/s, post-test speed 2.0 mm/s, target mode: strain, strain intensity: 35%, time: 5.0 s, trigger force 5.0 g, each sample was analyzed 3 times in parallel. The hardness, brittleness, viscosity, elasticity, cohesion, adhesion, chewing, and recovery properties were determined.

### 2.7. BAs Analysis

#### 2.7.1. Preparation of Standard BA Solutions

BA analysis was performed according to the method described by Jin et al. [18] with a minor modification. The standard substances of BAs, tryptamine, phenethylamine, putrescine, cadaverine, histamine, tyramine, spermidine, and spermine, were purchased from Sigma-Aldrich Sigma Chemical Co. (St. Louis, MO, USA). Each was dissolved in 0.1 M hydrochloric acid to prepare individual stock solutions at a concentration of 100 μmol/mL. A suitable volume of each individual stock solution was then mixed to prepare a combined standard solution, at a concentration of 1 μmol/mL, for future use. Detection is performed according to the concentration gradient shown in Table 1.

#### 2.7.2. Derivatization of Standard Substances

1 mL of the diluent was aspirated from bottle 2B and discarded to clean the pipette tip. Then, another 1 mL of the diluent was taken from bottle 2B and added to bottle 2A, which contained the derivatization powder. This mixture was vortexed for 10 s to ensure complete dissolution and was set aside for later use. If the derivatization powder did not completely dissolve, bottle 2A can be heated at 55 °C until fully dissolved, with the heating process not exceeding 10 min. A total of 10 μL of the standard solution was pipetted into the bottom of the derivatization tube, followed by the addition of 70 μL of AccQ·Tag Ultra Borate buffer. The mixture was vortexed to ensure homogeneity. Then, 20 μL of AccQ·Tag reagent was added while continuing to vortex. The tube was sealed with a closure film and heated at 55 °C for a reaction time of 10 min. After the reaction, the mixture was cooled to room temperature and transferred to a clean sample vial.

#### 2.7.3. Standard Curves and Linear Regression Equations for BAs

The retention time of eight BAs was obtained by a linear test of eight biogenic amine standards mixed solutions at the concentrations of 1, 5, 10, 50, 100, 150, 200, 250, and 300 nmol/mL, respectively. The linear regression equations and correlation coefficients R^2^ were obtained with the concentration as the horizontal coordinate, and the peak area as the vertical coordinate, and the results are shown in Table 2. All of them were greater than 0.99, indicating that there was a strong correlation between the peak area and the biogenic amine content. The relative standard deviation (RSD) of QC (Quality Control) samples was utilized to assess the stability of the instrument during the detection process. Generally, an RSD value below 30% indicates a relatively stable detection system. The RSD values for the BAs in the sufu samples were all below 5%. This finding provides a significant assurance of the reliability and repeatability of the data. All eight BAs were eluted within 10 min, with excellent separation, indicating the feasibility of the elution program. Overall, the UHPLC-QE (UHPLC-ESI-QE-Orbitrap-MS) method demonstrated accuracy and reliability, making it suitable for the detection of BAs in Mouding sufu.

#### 2.7.4. Sufu Sample Derivatization

Accurately transfer an appropriate amount of Mouding sufu sample into a centrifuge tube and add an equal volume of methanol to the sample, vortex thoroughly to ensure complete mixing, and let stand at −20 °C for 30 min, centrifuge at 12000 rpm for 10 min and remove the supernatant. Add an equal volume of 0.2 M hydrochloric acid, vortex mix well, and extract at room temperature for 1 h. Centrifuge at 12,000 rpm for 10 min, extract the supernatant, and dilute to an appropriate multiple. Then, add 10 μL of the sample into a UHPLC vial and add 70 μL Borate buffer and 20 μL AccQ Tag reagent. Keep the reaction mixture at room temperature for 1 min, heat at 55 °C for 10 min, and inject 1 μL after cooling. Conduct an on-instrument detection, according to the concentration gradient shown in Table 1.

#### 2.7.5. UHPLC-QE Analysis

Chromatographic separation was performed using a Waters BEH C18 (50 × 2.1 mm, 1.7 μm) column (Vanquish, UPLC, Thermo, Norristown, PA, USA). The gradient elution system consisted of solvent A, which was ultrapure water containing 0.1% formic acid, and solvent B, which was acetonitrile containing 0.1% formic acid. The flow rate was set at 0.5 mL/min, and the column temperature was maintained at 55 °C. A sample volume of 1 μL was injected. The elution gradient was as follows: at 0 min, A/B (95:5, *v*/*v*); at 5.5 min, A/B (90:10, *v*/*v*); at 7.5 min, A/B (75:25, *v*/*v*); at 8 min, A/B (40:60, *v*/*v*); at 8.5 min, A/B (95:5, *v*/*v*); and at 13 min, A/B (95:5, *v*/*v*). Throughout the analysis, samples were maintained at 4 °C in an automatic sampler. To minimize the impact of fluctuations in instrument detection signals, samples were analyzed in a random order. Quality control (QC) samples were uniformly inserted into the sample analysis queue to monitor and evaluate the stability of the system and the reliability of the experimental data. Mass spectrometry conditions instrument: Q Exactive high-resolution mass spectrometry system (Thermo, Norristown, PA, USA); Ion Source: Electrospray ionization (ESI); Sheath Gas: 40 arb; Auxiliary Gas: 10 arb; Spray Voltage: +3000 V; Capillary Temperature: 350 °C; Ion Transfer Tube Temperature: 320 °C; Scan Mode: Full MS mode; Scan Range: m/z 150–700.

### 2.8. Data Processing and Statistical Analysis

For each trial, quality analyses were tripled. Data were statistically calculated using Excel. The Pearson correlation coefficient was employed for bivariate analysis to investigate the relationship between the BA content and various physicochemical properties and texture characteristics. Statistical significance was determined at the 0.05 level (*p* < 0.05) using SPSS 23 (SPSS, IBM Corp., Chicago, IL, USA). Graphs were plotted using Origin 2021, and the results were expressed in the form of standard deviation (SD).

## 3. Results

### 3.1. Analysis of Physicochemical Indicators Changes

Amino acid nitrogen content refers to the nitrogen present in the form of amino acids and serves as a key indicator for assessing the degree of fermentation in fermented foods. It is also a crucial parameter for evaluating the quality of sufu [19]. As shown in Figure 2, the amino acid nitrogen content in the white substrate is initially minimal, measuring only 0.021 g/100 g. However, it exhibits a progressively increasing trend during the initial fermentation process of Mouding sufu, rising significantly to 0.236 g/100 g by the end of the fermentation. This increase is primarily attributed to the breakdown of proteins into amino acids and polypeptides during the early stages of fermentation [20]. The accumulation of peptides occurs at a slower rate, due to the delayed activity of proteases. As the fermentation progresses, peptides are further hydrolyzed into amino acids, resulting in a continuous increase in amino acid nitrogen content [21]. The taste profile of sufu is significantly influenced by its total acidity, which contributes to a smoother texture and enhanced palatability. Total acidity in the curd primarily arises from amino acids, fatty acids, and organic acids, which are produced through the enzymatic degradation of proteins and fats during fermentation [22]. However, excessive total acidity can negatively impact the shelf life of sufu. As illustrated in Figure 2, the total acidity of the white substrate (tofu) is initially 2.084 g/kg. After the initial fermentation stage, the total acidity of the hairy substrate increases to 4.945 g/kg. Throughout the fermentation process, total acidity generally demonstrates an upwards trend, with a significant increase observed between 48 h and 72 h. This rise is primarily attributed to the metabolic activity of microorganisms, which generate various organic acids during fermentation. Specifically, during the initial fermentation stage of Mouding sufu, the increase in total acidity is predominantly driven by the metabolic activity of lactic acid bacteria, leading to the production of organic acids, such as lactic acid [23]. Consequently, the total acidity increases after fermentation.

The soluble protein content in the white substrate is initially 3.106 g/kg and increases to 4.228 g/kg in the hairy substrate after the initial fermentation stage. As the fermentation progresses, the overall water-soluble protein content demonstrates an upward trend. This increase is primarily attributed to the activity of microorganisms, predominantly *Mucor racemosus,* during the initial fermentation stage. These microorganisms continuously secrete proteases, which accumulate on the tofu substrate and facilitate the degradation of proteins [22]. The proteases hydrolyze the proteins in the tofu into smaller water-soluble peptides and amino acids, resulting in the observed increase in water-soluble protein content.

The moisture content during the initial fermentation stage of sufu plays a critical role in influencing the mycelium formation and the catalytic activity of various enzyme systems. These factors subsequently affect the stability of the fermentation substrate and the production of diverse flavor compounds. As shown in Figure 2, during the transition from the white substrate to the hairy substrate stage, the moisture content remains within a range of approximately 64% to 79%, which is conducive to the growth of *Mucor racemosus*. The white substrate exhibits the highest moisture content at 78.94%. Following the initial fermentation stage, moisture gradually seeps out of the substrate, leading to shrinkage and hardening. Consequently, the moisture content decreases to 63.74% after the formation of the hairy substrate. During the initial fermentation, the moisture content shows a gradual decline, primarily due to the growth and reproduction of microorganisms such as *Mucor racemosus*, which consume a portion of the available moisture. Additionally, the tofu substrate placed in a natural environment loses moisture through evaporation. Variations in moisture content may also be attributed to differences in the gel networks within the substrate particles [24].

The pH of the white substrate is initially 5.83. After the initial fermentation stage, the pH of the hairy substrate increases to 7.22. The pH value exhibits a dynamic trend, initially decreasing and subsequently increasing over time. Notably, a significant rise in pH is observed during the transition from the white substrate to the hairy substrate. From the inoculation of *Mucor racemosus* to 24 h of fermentation, the pH value slightly decreases. This slight decrease might be due to the fact that during this stage, due to the hydrolysis of tofu components and the microbial fermentation of carbohydrates, free fatty acids, amino acids and peptides containing carboxyl side chains accumulate [3]. The pH decline is primarily caused by the accumulation of free amino acids resulting from rapid fermentation, which imparts a slightly sour taste to the tofu. Additionally, some moisture is lost from the tofu during this stage.

After 24 h of fermentation, the pH value continues to increase. This rise can be attributed to the progressive breakdown of proteins into amino acids, followed by deamination processes that generate ammonia [25], significantly elevating the pH. Furthermore, microorganisms may enter a stable growth phase, during which the rapid proliferation of *Mucor racemosus* results in a substantial depletion of carbon and nitrogen elements from the tofu substrate. The accumulation of ammonia compounds progressively contributes to the observed pH increase.

### 3.2. Texture Analysis

As shown in Table 3, the hardness of Mouding sufu initially increases and then decreases during the pre-fermentation process; at 48 h of fermentation, the hardness reaches its peak at 871.6 g. This increase can be attributed to the growth of microbial hyphae, which tightly envelop the tofu substrate and partially consume its water content, resulting in a gradual increase in hardness. After 48 h, the hardness decreases, due to further fermentation processes in which the microorganisms hydrolyze soybean proteins into smaller peptides and soluble proteins [24], softening the tofu substrate and reducing its hardness. The adhesiveness of the tofu increases steadily throughout the pre-fermentation period. During the white substrate stage, proteins coagulate to form a gel network structure, which provides strong water retention and results in lower adhesiveness, as the pre-fermentation progresses, enzymes produced by microorganisms degrade some macromolecules into smaller molecules, increasing the number of free substances and thereby enhancing the adhesiveness [3]. The elasticity of tofu generally demonstrates a decreasing trend during the pre-fermentation period, Prior to the microbial cultivation stage, the tofu substrate retains a high water content, contributing to its initially high elasticity. During the hairy substrate stage, the formation of a furry tofu layer covers the entire substrate and consumes a portion of the water. Concurrently, as soybean proteins undergo further degradation, the gel network structure of the hairy substrate becomes disrupted, resulting in a significant reduction in elasticity.

Cohesiveness represents the strength of the internal bonds within the sample, reflecting its resistance to damage and its ability to maintain structural integrity [25]. Studies have demonstrated that cohesiveness is associated with the protein structure or cross-linking within the sample [26]. During the pre-fermentation stage of sufu, the white substrate exhibits the highest cohesiveness, reaching 0.78 g, which indicates the tightest internal bonding at this stage. As the cultivation time progresses, cohesiveness decreases due to the gradual breakdown of proteins, which leads to the weakening of internal bonds. By the end of the pre-fermentation phase, the cohesiveness of the hairy substrate declines to 0.62 g. Gumminess, defined as the product of hardness and cohesiveness [27], exhibits a trend similar to that of hardness, initially increasing and subsequently decreasing. During the white substrate stage, gumminess remains low due to the protein coagulation. In the early stage of cultivation, the growth of *Mucor racemosus* on the substrate surface contributes to an increase in gumminess. However, as the pre-fermentation progresses, protein degradation leads to a continuous decrease in hardness, resulting in a reduction in gumminess. Chewiness, which is the product of gumminess and elasticity, also exhibits a trend that initially increases and subsequently decreases, closely mirroring the behavior of gumminess. The measurement of chewiness, as determined by a texture analyzer, is based on the physical texture properties of the food material [28], and variations in chewiness are predominantly driven by changes in hardness. Consequently, the trend in chewiness closely aligns with that of hardness.

In summary, during the pre-fermentation period of Mouding sufu, the textural properties such as hardness, adhesiveness, elasticity, cohesiveness, gumminess, and chewiness undergo significant changes. These changes are crucial for understanding and optimizing the fermentation process, ensuring the production of high-quality sufu.

### 3.3. Biogenic Amine Analysis

Using ultra-high-performance liquid chromatography coupled with a high resolution Orbitrap mass spectrometer (UHPLC-QE, Thermo, Norristown, PA, USA), samples were collected at five fermentation time points (0, 24, 48, 72, 96 h) to analyze the changes in biogenic amine (BA) content during the pre-fermentation stage of Mouding sufu. The results, as shown in Figure 3, indicate that seven out of the eight targeted BAs were detected, including tryptamine, phenylethylamine, spermine, tyramine, agmatine, putrescine, and cadaverine. Notably, histamine, which is considered the most toxic amine commonly found in food, was not detected throughout the entire pre-fermentation stage. The absence of histamine suggests that the tofu maintains good quality during this stage of fermentation [26].

Biogenic amines (BAs) are formed through the decarboxylation of free amino acids produced by microbial activity in food [27], During the pre-fermentation process of sufu, the proteins are degraded by microorganisms into nitrogen-containing compounds and amino acids, which serve as precursors for the biosynthesis of BAs, thereby influencing the dynamic changes in biogenic amine content [28]. The total biogenic amine content during pre-fermentation ranged from 60.66 to 38.19 mg/kg, following the order: 0 h > 24 h > 72 h > 48 h > 96 h. Among the detected BAs, spermidine exhibited the highest content, ranging from 16.63 to 38.23 mg/kg, followed by spermine (6.85 to 13.65 mg/kg) and putrescine (3.83 to 8.92 mg/kg). Different types of BAs displayed distinct trends during the fermentation process, as shown in Figure 3.

Spermidine, the predominant biogenic amine at the end of pre-fermentation, demonstrated a consistent decreasing trend throughout the process. In contrast, putrescine content initially increased, peaking at 72 h with a concentration of 10.37 mg/kg, before subsequently decreasing, yet showing an overall increase. Cadaverine exhibited minimal fluctuations, with its content decreasing from 3.34 mg/kg in the white tofu substrate to 2.36 mg/kg by the end of pre-fermentation. Tyramine content increased significantly during pre-fermentation with a more than 50-fold increase after 96 h of fermentation, compared to its initial level in the white tofu substrate. Both spermine and spermidine displayed a consistent decreasing trend throughout the pre-fermentation period. Phenylethylamine and tryptamine were present in relatively low concentrations, with little overall change observed. The maximum content of phenylethylamine was 0.96 mg/kg, while tryptamine reached a peak concentration of 0.60 mg/kg.

The declining trend in biogenic amine (BAs) content during the pre-fermentation stage is closely associated with changes in monoamine oxidase (MAO) activity [29]. This enzyme catalyzes the oxidation of BAs into acetaldehyde and ammonia, resulting in a reduction in their content.

### 3.4. Correlation Analysis Among Various Indicators

The correlation among the three factors is clearly illustrated in Figure 4A. Hardness, chewability, and adhesiveness exhibit significant positive correlation with texture, while no significant correlation is observed with other physicochemical indexes and BAs. A significant negative correlation is observed between recovery and soluble protein content. Elasticity shows a positive correlation with water content and a negative correlation with total acidity, amino acid nitrogen, soluble protein, and pH. Amino acid nitrogen is significantly positively correlated with pH, total acidity, and soluble protein content. Similarly, soluble protein is significantly positively correlated with amino acid nitrogen and total acidity but negatively correlated with water content and elasticity, consistent with the results discussed earlier. Spermine is positively correlated with water content and elasticity but negatively correlated with amino acid nitrogen, pH, total acid, and soluble protein. Tyramine exhibits a significant positive correlation with amino acid nitrogen, total acid, and pH, and significant negative correlation with water activity and elasticity, spermidine demonstrates significant positive correlation with the water content, and significant negative correlation with the soluble protein. Phenylethylamine, tryptamine, cadaverine, and putrescine show no significant correlations with structural or physicochemical indices. In Figure 4B, redundancy analysis (RDA) is employed to analyze the correlation among physicochemical indices during the pre-fermentation stage. The length of the arrows represents the strength of the correlation, with longer arrows indicating stronger correlations. Similarly, closer distances between variables indicate stronger relationships. Figure 4B highlights the correlation among the physicochemical indices during the pre-fermentation process of Mouding sufu. Water content exhibits a strong negative correlation with other physicochemical indices, aligning with the earlier analysis of trends in physicochemical changes during pre-fermentation. Total acidity shows a strong positive correlation with soluble protein, pH, and amino acid nitrogen. The increase in pH is attributed to the alkaline nature of amines and ammonia produced during the degradation of soluble protein and amino acids. Figure 4C presents the results of one-way ANOVA conducted on the first six central indicators, Significant differences are observed in the core indicators at each quality inspection stage, displaying a trend of initial decline followed by a rebound, consistent with the RDA results.

The physicochemical properties of Mouding sufu exhibited distinct variation patterns during different stages of pre-fermentation. Moisture content and pH generally demonstrated a decreasing trend, whereas amino nitrogen, total acidity, and water-soluble protein content increased as the fermentation progressed. Correlation analysis among these physicochemical indicators revealed that moisture content was negatively correlated with amino nitrogen, total acidity, and soluble protein. Meanwhile, pH was positively correlated with amino nitrogen and total acidity. Texture analysis of sufu at various pre-fermentation stages indicated specific changes: hardness, chewiness, and adhesiveness increased initially and then decreased, while springiness, cohesiveness, and resilience declined as the fermentation time advanced. These patterns were consistent with the correlations among texture indicators, and aligned with the findings reported by Kim Joo-Shin [30].

Spermine and spermidine exhibited a decreasing trend throughout pre-fermentation; however, their concentrations at the end of this stage remained significantly higher than those in the white tofu base. Similarly, putrescine content decreased but remained significantly elevated, compared to the white tofu substrate. Correlation analysis between physicochemical indicators and BAs revealed that tyramine was significantly positively correlated with pH, total acidity, and amino nitrogen, while spermine was significantly negatively correlated with these same factors. Furthermore, spermine and tyramine demonstrated a significant negative correlation, suggesting a potential interconversion between these two BAs during fermentation. The primary physicochemical factors influencing the formation and transformation of BAs were identified as pH, total acidity, and amino nitrogen. These findings provide new insights into the dynamics of BAs during the pre-fermentation stage of Mouding sufu and establish a foundation for understanding the interplay between physicochemical factors and microbial activity in this unique product.

### 3.5. Mechanism of Biogenic Amine Formation

Through the analysis of changes and correlations between biogenic amines (BAs) and physicochemical components in this study, a mechanistic diagram of the BAs production process based on physicochemical indicators is proposed, as illustrated in Figure 5. In food, BAs originate from the presence of free amino acids, microorganisms capable of producing amino acid decarboxylase, and environmental conditions conducive to BAs synthesis. During fermentation, enzymatic action degrades soy proteins in the tofu substrate into peptides and free amino acids, while microbial growth and metabolism also contribute to amino acid production. Arginine can undergo hydrolysis or decarboxylation to form ornithine or citrulline, which serve as important precursors for putrescine synthesis [31]. Putrescine can be further converted into spermidine and spermine [32]. Free amino acids are transformed into BAs through the action of amino acid decarboxylases. For instance, tyrosine decarboxylase converts tyrosine and phenylalanine into tyramine and phenylethylamine, respectively [33]. Similarly, other amino acids are converted into their corresponding BAs by specific amino acid decarboxylases.

## 4. Discussion

To investigate the changes and the influencing factors of BAs during the pre-fermentation process of Mouding sufu, this study utilized high-performance liquid chromatography-tandem mass spectrometry (HPLC-MS/MS) to monitor the dynamic changes in BA content. The results revealed that seven BAs were identified throughout the pre-fermentation stage of Mouding sufu tryptamine, phenylethylamine, spermidine, tyramine, spermine, cadaverine, and putrescine, while histamine was not detected. Among these, spermine, spermidine, and putrescine were identified as the predominant BAs.

The total biogenic amine (BAs) content exhibited a distinct temporal trend: 0 h > 24 h > 72 h > 48 h > 96 h. Putrescine content initially increased, peaking at 72 h, before subsequently decreasing; however, its overall content demonstrated an upward trend. Tyramine, although present in trace amounts, increased more than 50-fold by the end of the pre-fermentation process compared to its initial level in the white tofu substrate. In contrast, spermidine and spermine exhibited a consistent downward trend throughout the pre-fermentation stage. Cadaverine, phenylethylamine, and tryptamine were present at relatively low concentrations and displayed negligible changes during the pre-fermentation process. Correlation analysis revealed significant relationships between BAs and the physicochemical properties and texture characteristics of Mouding sufu. During pre-fermentation, the texture of the tofu was primarily influenced by amino acid nitrogen, pH, moisture content, and soluble protein content. Tyramine demonstrated significant positive correlations with pH, total acidity, and amino acid nitrogen, whereas spermine showed significant negative correlations with these same factors. Spermidine was significantly negatively correlated with soluble protein. Additionally, a significant negative correlation was observed between spermine and tyramine during the pre-fermentation process, while tryptamine displayed significant positive correlations with both cadaverine and spermidine. Other BAs did not exhibit significant correlations with physicochemical properties. The physicochemical properties and texture characteristics were measured and analyzed at different stages of pre-fermentation to explore the correlations between BAs and these indicators. These findings provide valuable insights into the entire production process of Mouding sufu, aiming to enhance the safety of its consumption while maintaining and improving product quality. This study elucidates the dynamic changes of BAs during the pre-fermentation stage of Mouding sufu and highlights the role of physicochemical factors in shaping the texture and BA profiles of the product.

## 5. Innovations

This study monitored BAs at different time points during the pre-fermentation of Mouding sufu and analyzed their change patterns. These findings provide a theoretical foundation for controlling and reducing BAs levels during the pre-fermentation stage. Additionally, physicochemical indicators and texture characteristics were measured and analyzed at various stages of pre-fermentation. The correlations between BAs and these indicators were explored, offering a theoretical basis for optimizing the entire production process of Mouding sufu. This research aims to enhance the safety of Mouding sufu consumption while maintaining and improving product quality.

## Figures and Tables

**Figure 1 foods-14-00583-f001:**
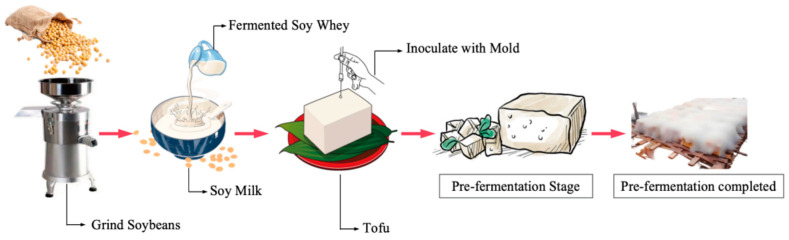
Sufu pre-fermentation process.

**Figure 2 foods-14-00583-f002:**
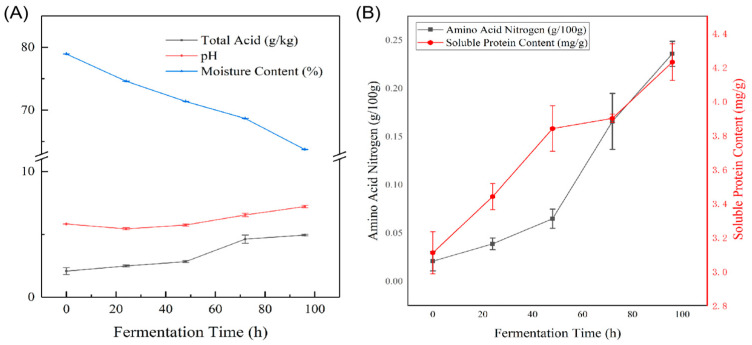
(**A**) Changes in total acid, pH, moisture content indicators during pre-fermentation. (**B**) Changes in indicators of amino acid nitrogen and soluble protein content during pre-fermentation.

**Figure 3 foods-14-00583-f003:**
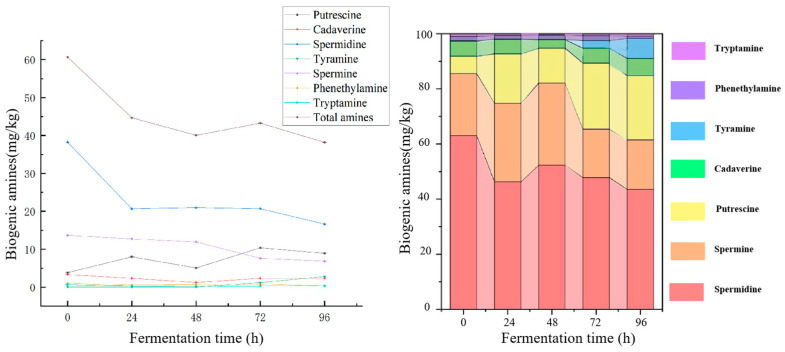
Biogenic amines (BAs) content at each stage of pre-fermentation.

**Figure 4 foods-14-00583-f004:**
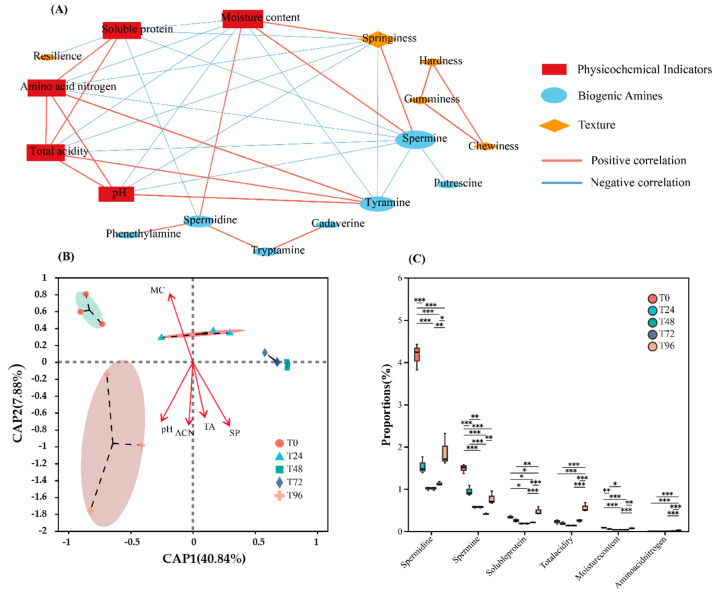
(**A**) Three-factor network analysis was conducted using physical and chemical indicators, biological amines and texture, and significant correlations were screened out. A network diagram was drawn to find that physical and chemical indicators and individual biological amines played a dominant role in the whole fermentation process and were correlated with other indicators. (**B**) RDA analysis was performed on the sample data constructed by physical and chemical indicators as environmental factors and other factors. (**C**) In the network analysis, the first six central indicators were screened for one-way ANOVA, and it was found that there were significant differences in the core indicators at each stage of quality inspection, and they showed a trend of gradual decline in the early stage and then rebound, consistent with the results of RDA. * *p* < 0.05, ** *p* < 0.01, *** *p* < 0.001.

**Figure 5 foods-14-00583-f005:**
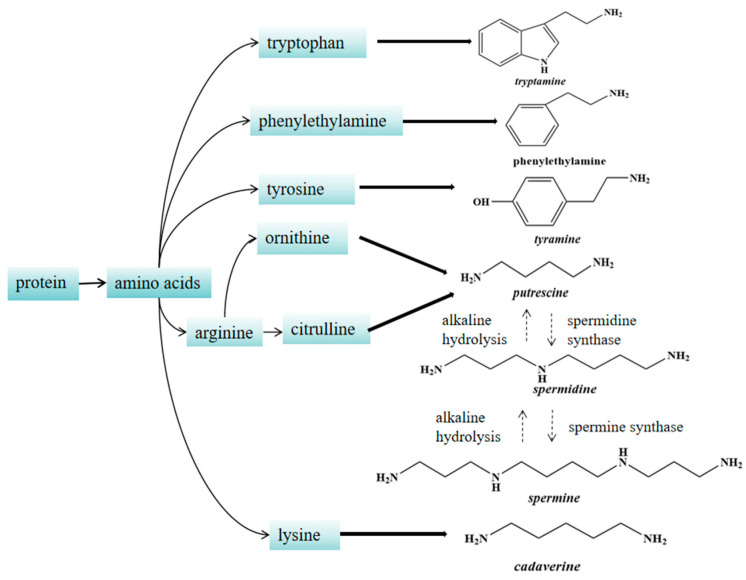
Mechanism of BAs Synthesis.

**Table 1 foods-14-00583-t001:** Information on mixed concentration gradient.

Standard Product	Concentration (nmol/mL)								
Putrescine	1	5	10	50	100	150	200	250	300
Cadaverine	1	5	10	50	100	150	200	250	300
Phenethylamine	1	5	10	50	100	150	200	250	300
Tyramine	1	5	10	50	100	150	200	250	300
Spermine	1	5	10	50	100	150	200	250	300
Spermidine	1	5	10	50	100	150	200	250	300
Tryptamine	1	5	10	50	100	150	200	250	300
Histamine	1	5	10	50	100	150	200	250	300

**Table 2 foods-14-00583-t002:** Retention time, linear equations, and R^2^ of BAs.

Biogenic Amine	Retention Period/min	Linear Equation	Correlation Coefficient R^2^
Histamine	0.72	Y = 1.784 × 10^5^X	0.9970
Cadaverine	6.43	Y = 6.189 × 10^4^X	0.9986
Spermidine	7.12	Y = 2.238 × 10^6^X	0.9993
Spermine	7.96	Y = 4.153 × 10^5^X	0.9981
Phenethylamie	9.31	Y = 1.180 × 10^7^X	0.9941
Tryptamine	9.32	Y = 7.566 × 10^6^X	0.9955
Putrescine	5.41	Y = 3.135 × 10^4^X	0.9956
Tyramine	7.17	Y = 7.302 × 10^6^X	0.9978

**Table 3 foods-14-00583-t003:** Changes of texture of Mouding fermented bean curd in the pre-fermentation period.

Fermentation Time (h)	Hardness(g)	Adhesiveness (g)	Springiness (g)	Cohesiveness (g)	Gumminess (g)	Chewiness (g)	Resilience (g)
0	338.8 ± 25.42c	−11.15 ± 0.85a	0.91 ± 0.01a	0.78 ± 0.03a	263.03 ± 17.01c	238.67 ± 16.43c	0.39 ± 0.02a
24	530.5 ± 50.61b	−10.65 ± 2.17bc	0.93 ± 0.03a	0.74 ± 0.02a	392.26 ± 43.57b	363.01 ± 41.04b	0.36 ± 0.02ab
48	871.6 ± 16.19a	−7.37 ± 0.71a	0.87 ± 0.01a	0.7 ± 0.05a	606.96 ± 32.52a	529.76 ± 37.14a	0.33 ± 0.04ab
72	802.21 ± 54.2a	−6.8 ± 1.25c	0.8 ± 0.03b	0.67 ± 0.06a	534.06 ± 11.16a	427.19 ± 13.85b	0.32 ± 0.03bc
96	416.46 ± 56.2c	−4.52 ± 1.98ab	0.72 ± 0.02c	0.62 ± 0.16a	260.33 ± 78.24c	164.29 ± 48.7c	0.26 ± 0.01c

Note: Different letters in the same column indicate significant differences (*p* < 0.05).

## Data Availability

The original contributions presented in the study are included in the article, further inquiries can be directed to the corresponding authors.

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
