# Peer review of "The Fermentation Law of Biogenic Amines in the Pre-Fermentation Process Is Revealed by Correlation Analysis"

_foods, 2025, doi:10.3390/foods14040583_

Round 1

Reviewer 1 Report

Comments and Suggestions for Authors

Dear Authors,

The manuscript provides valuable insights, but several aspects require clarification and correction:

1) The text formatting in some sections should be revised for consistency;

2) In Table 1, the classification of "Amphetamine" and "Necrotinamide" as biogenic amines needs verification. Similarly, it should be clarified whether "Phenethylamine" is a biogenic amine (Table 3);

3) Spelling errors in Table 1 and Table 2 should be corrected;

4) The chromatographic column's name needs to be specified more precisely;

5) In Section 2.9, the sample preparation method should be clarified should it follow the same derivatization process as the standards?

6) The legend in Figure 3 should be improved to clearly display the names of the analyzed compounds.

These revisions would enhance the clarity, accuracy, and overall quality of the article.

Author Response

Reviewer #1:

The manuscript provides valuable insights, but several aspects require clarification and correction:

1. The text formatting in some sections should be revised for consistency.

Response: We sincerely thank the reviewers for their meticulous review and valuable feedback. We have thoroughly revised the manuscript in accordance with the journal's template requirements, ensuring uniformity in line spacing, font size, font style, and paragraph indentation throughout the text.

2. In Table 1, the classification of "Amphetamine" and "Necrotinamide" as biogenic amines needs verification. Similarly, it should be clarified whether "Phenethylamine" is a biogenic amine (Table 3).

Response: We greatly appreciate the reviewer’s comments. The spelling errors of “Amphetamine” and “Necrotinamide” identified in Table 1 and Table 3 have been noted, and the necessary corrections have been made in the text. A comprehensive review of the entire manuscript has been conducted to ensure accuracy. The correct terms are “Cadaverine” and “Phenethylamine”, both of which are recognized biogenic amines, as illustrated in Figure 3 and section 3.3. Furthermore, we have revised the gradient elution program (Table 2) and incorporated this information as a textual statement in section 2.7.6. The pertinent text has been highlighted in red for clarity.

Page 5, line 206: The gradient elution system comprised solvent A, which consisted of ultrapure water with 0.1% formic acid, and solvent B, which was acetonitrile also containing 0.1% formic acid. The flow rate was maintained at 0.5 mL/min, and the column temperature was set at 55 °C. A sample volume of 1 μL was introduced for analysis. The elution gradient was as follows: at 0 min, A/B (95:5, v/v); at 5.5 min, A/B (90:10, v/v); at 7.5 min, A/B (75:25, v/v); at 8 min, A/B (40:60, v/v); at 8.5 min, A/B (95:5, v/v); and at 13 min, A/B (95:5, v/v). Throughout the analysis, samples were maintained at 4 °C in an automatic sampler.

3. Spellingerrors in Table 1 and Table 2 should be corrected.

Response: Thank you very much for pointing out the errors. The entries “Amphetamine” and “Necrotinamide” in Table 1(or table 3) is spelling errors. We have modified this error and highlighted it in red in the main text.

Page 4, line 191:

Table 1. Information of mixed concentration gradient.

Standard product

Concentration(nmol/mL)

Putrescine

1

5

10

50

100

150

200

250

300

Cadaverine

1

5

10

50

100

150

200

250

300

Phenethylamine

1

5

10

50

100

150

200

250

300

Tyramine

1

5

10

50

100

150

200

250

300

Spermine

1

5

10

50

100

150

200

250

300

Spermidine

1

5

10

50

100

150

200

250

300

Tryptamine

1

5

10

50

100

150

200

250

300

Histamine

1

5

10

50

100

150

200

250

300

Page 4, line 192:

Table 2. Retention time, linear equations, and R2 of BAs.

Biogenic amine

Retention period/min

Linear equation

Correlation coefficient R2

Histamine

0.72

Y=1.784e5X

0.9970

Cadaverine

6.43

Y=6.189e4X

0.9986

Spermidine

7.12

Y=2.238e6X

0.9993

Spermine

7.96

Y=4.153e5X

0.9981

Phenethylamie

9.31

Y=1.180e7X

0.9941

Tryptamine

9.32

Y=7.566e6X

0.9955

Putrescine

5.41

Y=3.135e4X

0.9956

Tyramine

7.17

Y=7.302e6X

0.9978

4. The chromatographic column's name needs to be specified more precisel.

Response: Thank you very much for the reviewer’s comments. 

We have added corresponding descriptions in the main text. The chromatography column we used was a Waters BEH C18 (50 × 2.1 mm, 1.7 μm).

Mobile phase: A phase is ultrapure water solution (containing 0.1% formic acid), B phase is acetonitrile solution (containing 0.1% formic acid); Flow rate: 0.5 mL/min; Column temperature: 55 °C; Injection volume: 1 μL; Elution gradient: 0 min A/B (95:5, v/v), 5.5 min A/B (90:10, v/v), 7.5 min A/B (75:25, v/v), 8 min A/B (40:60, v/v), 8.5 min A/B (95:5, v/v), 13 min A/B (95:5, v/v). During the entire analysis process, the samples were kept in an automatic sampler at 4 °C.

Page 5, line 205: Chromatographic separation was performed using a Waters BEH C18 (50 × 2.1 mm, 1.7 μm) column (Vanquish, UPLC, Thermo, USA). The gradient elution system consisted of solvent A, which was ultrapure water containing 0.1% formic acid, and solvent B, which was......were uniformly inserted into the sample analysis queue to monitor and evaluate the stability of the system and the reliability of the experimental data.

5. In Section 2.9, the sample preparation method should be clarified should it follow the same derivatization process as the standards?

Response: We sincerely appreciate the reviewers for their careful review. We have added a description of the derivatization of sufu samples in the text, which have the same derivatization process with the standard substances. As depicted in the 2.7.5.

Page 5, line195: Accurately transfer......acid solution. Vortex to mix again and extract at room temperature for 1 h. Centrifuge at 12000 rpm for 10 min, collect the supernatant, and dilute to an appropriate factor. Transfer 10 μL of the sample into a derivatization tube, add 70 μL of AccQ•Tag Ultra Borate buffer and 20 μL of AccQ•Tag reagent, vortex to mix, then heat at 55 °C for 10 min. After cooling, carry out the detection on the machine.Conduct on-instrument detection according to the concentration gradient shown in Table 1.

6. The legend in Figure 3 should be improved to clearly display the names of the analyzed compounds.

Response: Thank you very much for the reviewer’s recommendations. The legend in Figure 3 has been improved for clearer visibility of the names of each compound. Additionally, we have enhanced the clarity of all the images in the paper for better presentation of the experimental results.

Reviewer 2 Report

Comments and Suggestions for Authors

Line 32-33: Word "fermented" is duplicated

Line 51 and 55 : Mucor change to  italics. Please check  throughout the manuscript that the genera and species are written in italics.

Line 60: what does BAs mean? Please, write the complete word when it first appears in the manuscript.

Line 74: Chen et al., "C" in capital letter

Line 85: "moisture  content" is repeated.

Methodology section 2.1: The title mentions the "Critical control points", What are these critical control points? Have they been identified using HACPP?

Line 182: "HCL" change to HCl

Figure 3: One of the graphs does not allow the information in the other image to be clearly seen

Homogenize the format for hours, in some cases it appears as "h" and in others as "hour"

2.1. Fermented Bean Curd Production Process and Critical Control Points

2.1. Fermented Bean Curd Production Process and Critical Control Points

The manuscript has a large number of grammatical errors, it is recommended to review it carefully. There are words where letters are missing or the order of the letters is changed.

Author Response

1. Line 32-33: Word "fermented" is duplicated; Line 51 and 55 : Mucor change to italics. Please check throughout the manuscript that the genera and species are written in italics;Line 60: what does BAs mean? Please, write the complete word when it first appears in the manuscript; Line 74: Chen et al., C in capital letter; Line 85: moisture content is repeated; Line 182: HCLchange to HCl.

Response: We would like to express our sincere gratitude to the reviewers for their meticulous evaluation and insightful comments, which have greatly enhanced the quality of our manuscript. We have revised the manuscript in accordance with the journal's template requirements, ensuring uniformity in line spacing, font size, font style, and paragraph indentation. Additionally, we have addressed the specific issues raised by the reviewers, including correcting spelling errors and standardizing unit abbreviations, all formatted consistently according to the journal's guidelines. We appreciate the time and effort invested in reviewing our manuscript and providing constructive feedback, which has significantly improved our work, such as:

Page 2, line 48: During the pre-fermentation process, tofu is inoculated with Mucor racemosus on the surface and cultivated for 96 h until the formation of furry tofu blocks occurs.

Page 2, line 50: Subsequently, the prepared furry tofu block is dried for approximately 12 h before being cured for 2–3 days.

Page 2, line 88: Select high-quality soybeans that are plump, free of insect damage, and mold-free. After cleaning, soak the soybeans for 8 h to 18 h until they are completely softened.

Page 3, line 104: Measure an appropriate amount of mold liquid medium into an Erlenmeyer flask according to the required volume of spore suspension. Add deionized water in proportion, seal the flask, and sterilize it in an autoclave at 120 ℃ for 20 min.

Page 3, line 109: Transfer the cultured medium into centrifuge tubes and centrifuge for 10 min. 

Page 3, line 115: Select 200 g of plump, mold-free soybeans per batch (one batch equals one tofu block). Soak the soybeans in an appropriate amount of deionized water at room temperature for 12 h to 18 h.

Page 3, line 124: Place the fermentation boxes in a constant temperature and humidity incubator, maintaining a relative humidity of 75% to 85% and a temperature of 20 â„ƒ to 25 â„ƒ for 4 d. 0, 24, 48, 72, 96 h of fermentation.

Page 4, line 148: The specific parameters of Texture Profile Analysis (TPA) were ...... time: 5.0 s, trigger force 5.0 g, each sample was done 3 times in parallel.

Page 6, line 276: From the inoculation of Mucor racemosus us to 24 h of fermentation, the pH value slightly decreases.

Page 6, line 277: After 24 h of fermentation, the pH value continues to increase.

Page 7, line293: As shown in Table 3, the hardness of Mouding sufu initially increases and then decreases during the pre-fermentation process, at 48 h of fermentation...... into smaller peptides and water-soluble proteins.

Page 1, Line 28-29: Sufu, often referred to as “Chinese cheese”[1], is a traditional Chinese fermented soybean product with distinct cultural characteristics.

Page 1, Line 45 and 49: Mucor racemosus appears to be the most widely employed strain by the manufacturers in Mouding for sufu production. According to traditional fermentation techniques of sufu, the production process comprises pre-fermentation and post-fermentation stages. During the pre-fermentation process, tofu is inoculated with Mucor racemosus on the surface and cultivated for 96 h until the formation of furry tofu blocks occurs.

Page 2, Line 53: The content of biogenic amines(BAs), hardness, and smoothness are important indicators for evaluating the quality and acceptability of fermented bean curd, as well as key factors in determining the maturity stage during the production process.

Page 2, Line 65: For instance, Cheng et al[11] investigated biochemical changes related to protein degradation in low-temperature fermented bean curd, suggesting that Bacillus flavus could serve as a substitute strain for production.

Page 2, Line 74: It investigates key physicochemical properties such as moisture content, water-soluble proteins, total acidity and amino acid nitrogen, while tracking and detecting the types and levels of biogenic amines throughout pre-fermentation.(The repeated phrase "moisture content" has been removed from the sentence.)

Page 3, Line 122: Inoculate the tofu blocks by immersing all six sides evenly in a mold spore suspension. Arrange the tofu blocks neatly in fermentation boxes, leaving some space around each block to allow for Mucor racemosus growth, which promotes better fermentation[7].

Page 3, Line 124: Place the fermentation boxes in a constant temperature and humidity incubator, maintaining a relative humidity of 75% to 85% and a temperature of 20 ℃ to 25 ℃ for 4 d. 0, 24, 48, 72, 96 h of fermentation. At each time point, collect 7 samples, tracking three batches in parallel.

Page 6, Line 272: From the inoculation of Rhizop us to 24 h of fermentation, the pH value slightly decreases. This slight decrease might be due to the fact that during this stage, Due to......

Page 6, Line 278: After 24 h of fermentation, the pH value continues to increase.

Page 9, Line 379: The figure shows the correlation between physical and chemical indexes in the pre-fermentation process of Mouding sufu.

2. Figure 3: One of the graphs does not allow the information in the other image to be clearly seen.

Response: Thank you very much for the reviewer’s recommendations. The legend in Figure 3 has been improved for clearer visibility of the names of each compound. Additionally, we have enhanced the clarity of all the images in the paper for better presentation of the experimental results.

3. Methodology section 2.1: The title mentions the Critical control points, What are these critical control points? Have they been identified using HACPP?

Response: We sincerely appreciate the reviewers for their careful review. "Critical control points" are provided solely to illustrate the sample production process, offering a brief overview of the core procedural steps involved in the production of fermented tofu prior to fermentation, and should not be confused with HACCP. We have made more accurate descriptions in the text and highlighted them in red.

Page 2, Line 99: Mix the cleaned tofu blocks with salt, chili powder, and spices. Place the mixed tofu blocks into fermentation containers, stored for around 180 days, this stage is referred to as the post-fermentation stage. Figure 1 illustrates the critical control points of the sufu pre-fermentation process.

4. The manuscript has a large number of grammatical errors, it is recommended to review it carefully. There are words where letters are missing or the order of the letters is changed.

Response: We sincerely appreciate the reviewers for their careful review. We have conducted a comprehensive and detailed review of the entire manuscript and have carefully revised errors in spelling, grammar, and inconsistencies in capitalization, such as:

Page 5, line 226: Amino acid nitrogen content refers to the nitrogen present in the form of amino acids and serves as a key indicator for assessing the degree of fermentation in fermented foods. It is also a crucial parameter for evaluating the quality of sufu [18]. As shown in Figure 2, the amino acid nitrogen content in the white substrate is initially minimal, measuring only 0.021g/100g. 

Page 5, line 230: However, it exhibits a progressively increasing trend during the initial fermentation process of Mouding sofu, rising significantly to 0.236 g/100g by the end of fermentation. This increase is primarily attributed to the breakdown of proteins into amino acids and polypeptides during the early stages of fermentation[19].

Page 5, line 233: The accumulation of peptides occurs at a slower rate, due to the delayed activity of proteases. As fermentation progresses, peptides are further hydrolyzed into amino acids, resulting in a continuous increase in amino acid nitrogen content[20].

Page 6, line 280: After 24 h of fermentation, the pH value continues to increase. This rise can be attributed to the progressive breakdown of proteins into amino acids, followed by deamination processes that generate ammonia[25], significantly elevating the pH. Furthermore, microorganisms may enter a stable growth phase, during which the rapid proliferation of Mucor racemosus results in a substantial depletion of carbon and nitrogen elements from the tofu substrate. The accumulation of ammonia compounds progressively contributes to the the observed pH increase.

Page 7, line 319: Chewiness, which is the product of gumminess and elasticity, also exhibits a trend that initially increases and subsequently decreases, closely mirroring the behavior of gumminess. The measurement of chewiness, as determined by a texture analyzer, is based on the physical texture properties of the food material[24], and variations in chewiness are predominantly driven by changes in hardness. Consequently, the trend in chewiness closely aligns with that of hardness.
